# Anti-Obesity Efficacy of *Pediococcus acidilactici* MNL5 in *Canorhabditis elegans* Gut Model

**DOI:** 10.3390/ijms23031276

**Published:** 2022-01-24

**Authors:** Kaliyan Barathikannan, Ramachandran Chelliah, Fazle Elahi, Akanksha Tyagi, Vijayalakshmi Selvakumar, Paul Agastian, Mariadhas Valan Arasu, Deog-Hawn Oh

**Affiliations:** 1Department of Food Science and Biotechnology, College of Agriculture and Life Science, Kangwon National University, Chuncheon 24341, Korea; bkannanbio@gmail.com (K.B.); ramachandran865@gmail.com (R.C.); elahidr@gmail.com (F.E.); akanksha.tyagi001@gmail.com (A.T.); vijiselva10@gmail.com (V.S.); 2Agricultural and Life Science Research Institute, Kangwon National University, Chuncheon 24341, Korea; 3Kangwon Institute of Inclusive Technology (KIIT), Kangwon National University, Chuncheon 24341, Korea; 4Department of Plant Biology and Biotechnology, Loyola College, Chennai 600 034, India; agastian@loyolacollege.edu; 5Department of Botany and Microbiology, College of Science, King Saud University, P.O. Box 2455, Riyadh 11451, Saudi Arabia; mvalanarasu@gmail.com

**Keywords:** *Pediococcus acidilactici*, cholesterol-lowering ability, lipid reduction, *C. elegans*, qPCR

## Abstract

In the present study, thirty two lactic acid bacteria (LAB) were isolated from fermented Indian herbal medicine. In comparison to other strains, MNL5 had stronger bile salt hydrolase (BSH) and cholesterol-lowering properties. Furthermore, it can withstand the extreme conditions found in the GI tract, due to, e.g., pepsin, bile salts, pancreatin, and acids. *Pediococcus acidilactici* MNL5 was identified as a probiotic candidate after sequencing the 16S rRNA gene. The antibacterial activity of *P. acidilactici* MNL5 cell-free supernatants (CFS) against *Escherichia coli*, *Staphylococcus aureus*, *Helicobacter pylori, Bacillus cereus*, and *Candida albicans* was moderate. A *Caenorhabditis elegans* experiment was also performed to assess the effectiveness of *P. acidilactici* MNL5 supplementation to increase life span compared to *E. coli* supplementation (DAF-2 and LIU1 models) (*p* < 0.05). An immense reduction of the lipid droplets of *C. elegans* was identified through a fluorescent microscope. The drastic alteration of the expression of fat genes is related to obesity phenotypes. Hence, several paths are evolutionary for *C. elegans*; the results of our work highlight the nematode as an important model for obesity.

## 1. Introduction

Probiotics are living microbial dietary supplements that impact the host by strengthening microbial gut stability [1]. Reports have documented that there is a discrepancy between information on the labels and data reported following laboratory safety assessments of various commercial probiotic products [2,3]. Lactic acid bacteria (LAB) such as *Lactococcus*, *Pediococcus*, *Enterococcus*, *Streptococcus*, *Weissella*, and *Saccharomyces* are common commercial probiotics. The primary criteria for selecting probiotic bacteria include hemolytic activity, antibiotic resistance, antimicrobial activity, pepsin and pancreatin tolerance, bile tolerance, morphologies, genotypes, and carbohydrate tolerance [4]. Different studies of LAB have noted a broad range of health-enhancing characteristics associated with microbial balance in the host gut flora [5]. In addition, probiotics have the ability to produce bile-salt hydrolase, which considerably decreases cholesterol levels [6,7]. Lactobacilli belong to the primary group known as the LAB that is Generally Recognized as Safe (GRAS) [8]. *Lactobacillus* and *Pediococcus* are able to maintain a healthy intestinal microbial ecosystem and reduce the function of cholesterol [9,10].

Furthermore, isolates showing unique features that could be beneficial in biotechnology or probiotic use could be produced by isolating and identifying new strains from un-investigated medicinal and edible plants [11,12]. Recently, *C. elegans* have been used in fat metabolism research. Bacterial dietary fatty acids are transformed into triglycerides, which are the main form of fat deposited on the epithelial and gut organs of *C. elegans*. Most of the key biosynthetic pathways of humans, such as synthesis with fatty acids, elongation, denaturation, and β-oxidation, as well as pathways for the neuropeptides [13], are well preserved in *C. elegans*. Fatty acid metabolism shows a vital feature of controlling stress resistance pathways, even without the activity of DAF-16, if nematode DAF-2/insulin signaling is regulated. This allows the identification of the specific aspects and pathways that act downstream of fat regulators through expression profiling and epistasis evaluation of potential metabolic genes [14]. Several studies have indicated that the consumption of LAB and other probiotics reduces coronary artery risk and the severity of symptoms of Bowel’s syndrome [15]. In our study, we identified and evaluated lactic acid bacteria based on BSH activity, cholesterol reduction capacity, probiotic capabilities, and food safety aspects. Additionally, the mechanism of action for inhibiting fat accumulation in a *C. elegans* gut model by *P. acidilactici* MNL5 was studied. This research underlines the ability of *P. acidilactici* MNL5 to reduce multitarget lipid lowing active substance to help in the management of obesity.

## 2. Results

### 2.1. Isolation and Identification of Cholesterol-Lowering LAB

Moola nivarana lehyam is a fermented herbal medicine, mainly applied for reducing obesity, which consist of combined dried spices such as *Medhya Rasayan* (Nootropic herb), *Terminalia chebula*: 3.10%, *Emblica officinalis*: 3.10%, *Terminalia bellirica*: 3.10%, *Withania somnifera* (6.85%), *Zingiber officinale* (3.6%), *Phytophthora megasperma* (root rot) (3.6%), *Cyperus rotundus* (7%), *Tamarindus indica* (7%), *Aloe barbadensis miller* (7%), *Cissus quadrangularis* (7%), *Amorphophallus paeoniifolius* (3%), *Arenga pinnata* (5%), butter (10%), where naturally fermented at 30 °C for 72 h. The lactic acid bacterial species have been isolated from fermented herbal medicine products, obtained from India. Total 32 LAB strains were examined for BSH and cholesterol assimilation (Table 1). Among them, six LAB strain expressed BSH activity and cholesterol assimilation ranged from 9.5 to 89.5%. The MNL5 had the highest cholesterol (89.5%) and BSH activity (16 mm zone of precipitation) out of 32 strains. The isolate MNL5 was compared with 16S rRNA gene sequences in GenBank and it was very similar (99% (MK928489)) with the *P. acidilactici* strain (Table 1, Figure 1 and Figure 2). *P. acidilactici* MNL5 strain content in moola nivarana lehyam 3.69 log CFU/g.

### 2.2. Antibiotic Resistance

All the isolates were tested with nine different antibiotics (Appendix A). Most of the bacterial isolates were susceptible with ampicillin, streptomycin, penicillin, tetracycline, kanamycin, erythromycin, gentamicin, clindamycin, and chloramphenicol. The *P. acidilactici* MNL5 was showed the resistance with nine antibiotics and the evaluation was according to the European Food Safety Authority. Hence, in this results supports that the consumption of *P. acidilactici* MNL5 increase a human health while administration with antibiotic.

### 2.3. Antimicrobial Activity

A broad-spectrum antimicrobial activity was observed in thirty-two isolates against gram positive and negative pathogens. Above all LAB, the MNL5 exhibits the substantial inhibitory activity against *Escherichia coli, Staphylococcus aureus*, *Bacillus cereus* and *H. pylori* while lesser activity observed with *Candida albicans* KCTC 7965 (Appendix A).

### 2.4. Hemolytic Activity

The tested bacteria showed γ-hemolytic (no zone effect), α-hemolysis (green zone), and none had β-hemolytic (blood lysis zone) effects (Table 1). Probiotic strain screening should consider safety issues, including specifications such as origin, identity, and lack of harmful activities [16].

### 2.5. Survival under In Vitro Gastrointestinal Conditions

#### Tolerance to Low pH, Pepsin and Bile Salts and Pancreatin

In our study, the survival of *P. acidilactici* MNL5 under in vitro gastrointestinal conditions like low pH (>94%), pepsin (>76%), bile salts, and pancreatin (>77%) was increased (Figure 3). According to the results, MNL5 showed significant resistance in simulated intestinal conditions.

### 2.6. Chemotaxis Assays

In our experiment, the nematodes showed minimal preferences for the MNL5 pellet and strong inclinations toward broth culture (liquid) cultivation. This finding may be due to the attraction of nematodes to the diacetyl group, which is one of the secondary metabolites produced by lactic acid bacteria. Based on the results, MNL5 appears to promote nematode growth when administered as a nematode supplement. We found that significantly more animals selected the MNL5 diet supplemented with sugar than the control diet.

### 2.7. Worm Size Measurement

According to our results, the body size of *C. elegans* altered significantly throughout the first three days of the diet. Body size was increased in the *E. coli* OP50+Glucose diet, while MNL5+Glucose was decreased compared to the regular *E. coli* OP50 diet. Our findings suggest that increasing the survival rate while decreasing the size of worms employed leads to a long life for both the DAF-2 and LIU1 models. (Appendix A).

### 2.8. C. elegans Gut Colonization Ability

The population of gut bacteria accumulated while administered MNL5+ Glucose and *E. coli* OP50 significantly diminished when compared to *E. coli* OP50+ Glucose diet colonization (Figure 4). These results indicate that early bacterial accumulation is associated with a shorter period of life for nematodes. We also pointed out that the increased lifespan of the worm depended on the DAF-2/DAF-16 function, and on the colonization of the bacterial strains in the gut of the worms if the MNL5 strain was presented as food sources to the worms. Nevertheless, our results suggest that these newly isolated MNL5 strains might have health-promoting impacts on the host.

### 2.9. C. elegans Life Span Assay

The findings revealed that P. acidilactici MNL5 has a long life span and protective nematode effects that benefit the DAF-2 and LIU1 models. The life span of *C. elegans* (DAF-2 and LIU1 model) increased supplemented with P. acidilactici MNL5 + Glucose (<32 ± 1 days; <27 ± 1), while compared to *E. coli* (29 ± 1; 24 ± 1) and *E. coli* OP50 + Glucose (26 ± 1; 17 ± 1) is decreased. (Figure 5). When comparing the control results observed with the test results, we found that the P. acidilactici MNL5 showed that the worms longer lifespan.

### 2.10. Lipid Determination by Microscope Analysis

Aging dysregulates intracellular lipid metabolism in several organisms. Experiments DAF-2 and LIU1 model showed that the Lipid droplets are decreased *P. acidilactici* MNL5 with Glucose supplement treatment on end day (28th and 25th day) of the experiment, similarly, the lipid droplets increased significantly after the OP50 with Glucose treatment by fluorescence microscopy analysis (Figure 6).

### 2.11. Transcriptomic Analysis in C. elegans

In our study, we performed a demonstration of DAF-2 and LIU1 with modified fat composition, examining the relationships between particular lipids and biological functions. The qPCR experiments suggested that the three delta-9 desaturase homologs, i.e., fat-4 fat-5 and fats-6, encoding fatty acid desaturase enzyme, encoding double bond formation in a saturated fatty acid metabolism related gene expression were decreased. The obtained standard curve for each target gene with its respective regression coefficient (R^2^ value) and efficiency (%) is represented in Appendix A. The developed qPCR assay was found to work efficiently over a cDNA concentration range of 1 × 10^2^, 1 × 10^1^, 1 × 10°, 1 × 10^−1^, 1 × 10^−2^, 1 × 10^−3^ ng per reaction. This experiments pathway was intended to demonstrate that the daf-2 and LIU1 model responds to the absence of lipids by either up- or down- regulating these pathways. Based on our qPCR analysis, we showed the active role of transcriptional upregulation of *E. coli* + glucose treated worms in increasing de novo fatty acid synthesis. However, *P. acidilactici* MNL5 + glucose treated worms indicated that the suppression of transcription of the fatty acid desaturase genes was downregulated. As a result, we showed that fat-4, fat-5, and fat-6 were modulated by a mechanism that suppresses lipid reduction by *P. acidilactici* MNL5 in a *C. elegans* fat gene model (Figure 7).

## 3. Discussion

In this study, lactic acid bacterial species were isolated from fermented herbal medicine. Nutritional foods are essential for the quality of life and survival of organisms. Probiotic bacteria play a significant role in digestion and survive in the gastrointestinal tract due to their acid tolerance in human gastric juices and bile in the gut. In this study, 32 LAB were identified from fermented herbal medicine products from India, and analyzed for BSH and cholesterol assimilation. Among them, *P. acidilactici* MNL 5 exhibited antibacterial activity with *Escherichia coli* ATCC 35150, *Staphylococcus aureus* ATCC 13,150, and *Bacillus cereus* ATCC 14,579. Additionally, some LAB strains were found to interact with a wide range of antimicrobial drugs. The functional activity of *P*. *acidilactici* in fermented foods leads to the production of phytase and lactic acid through homofermentation [17]. In addition, *P*. *acidilactici* was shown to enhance thep-hydroxy-benzoic, vanillic acids and crude fiber, but to reduce protein and crude fat content in fermented barley [18]. Furthermore, *P*. *acidilactici* enhanced the flavor and aroma characteristics of bioactive phenolic compounds (catechin, quercetin, hydroxybenzoic acids and myricetin) [19,20,21]. Pathogenic bacteria with elevated levels of antimicrobial activity are key effective probiotics [22]. Resistance to gastrointestinal juices throughout the intestinal tract and stomach is the main factor determining whether probiotic strains reach the gut and benefit the host [23]. Before confirming the efficiency of a probiotic bacterium, its biological impact on the host during intestinal passage must be tested [24]. The probiotic physiological activity should resist stress in digestive enzymes such as pepsin, pancreatin, bile salt, and acidic medium (pH 2.0) [6]. For example, if exposed to low pH, *L. delbrueckii* subsp and *L. bulgaricus* exhibit low survival rates [25]. In this study, the selected MNL5 strains showed good tolerance to low pH (>94%), pepsin (>76%) and bile salts and pancreatin (77%) (see Figure 3).

The nematode *C. elegans* is a useful in vivo host model to investigate the interactions among probiotic bacteria. Probiotics have a health-promoting effect recognitions to their ability to colonize the host gut; therefore, their adhesion and colonization abilities must be evaluated in studies of the potential benefits of their use [3,26]. The chemotaxis test was used to assess *C. elegans* odor reactions and their food preferences. Our experiment showed that these nematodes were disinclined to consume MNL5 pellets, instead favoring liquid culture. Additionally, the behavioral characteristics of nematodes clearly demonstrated, through a chemotaxis test, that the MNL5 strains are nontoxic. Based on these findings, it may be concluded that LAB plays a role in growth enhancement when used as a nematode food supplement. Previous studies indicated that some bacteria may colonize the intestines of worms, lengthening the lifespan of the nematode. *C. elegans* has been used as a model organism for over half a century, with its many similarities between mammals and microbes facilitating assessments of host–microbe interactions [27]. Coenzyme Q (CoQ), an essential coenzyme with aerobic respiration, plays a significant role in increasing *C. elegans* longevity. Since the lifespans of nematodes decrease following exposure to *E. coli*, it would appear that LAB also has a set of CoQ synthesizing enzymes that lead to lifespan extension. When LAB produces CoQ, *C. elegans* will have a longer lifetime [3,28]. In our study, the lifespans of DAF-2 and LIU1 decreased considerably due to lipid metabolism. The longevity of the DAF-2 and LIU1 models was significantly improved following exposure to *P. acidilactici* MNL5.

*C. elegans* extends the lifespan, and requires the activity, of DAF-16, FOXO longevity controlled transcription factors, known as IGF-I Signaling Pathways (IIS), in response to insulin/IGF. As several studies have shown, elevated glucose levels in *C. elegans* reduce their fertility and lifespan; indeed, the efficiency of regulating oxidation increases substantially with increasing exposure to glucose [29]. *C. elegans* has been used in studies of metabolism related diseases, such as the regulation of fat storage and obesity [30]. The transparency of *C. elegans* makes it easy to view the fluorescent Nile Red, a commonly used dye for staining and measurement of lipid stores within worm intestine [31]. In our experiment, on the 15th day of *P. acidilactici* MNL5 and glucose supplement treatment, we observed a significant reduction in lipid droplets after OP50 with glucose lipid droplets is increased (Figure 6). *Drosophila* leads, for example to, delayed growth and mutations in the insulin/IGF receptor, as well as and somatotropic signaling disruptions in young worms, leading to lower body size in mammals and reduced IGF-1 circulation [32]. Accordingly, fatty acid oxidation related to fat metabolism in mice is upregulated by *L. curvatus* HY7601, *L. plantarum* KY1032, and *B. breve* B-3 [22,33,34,35]. Thus, our results support the hypothesis that the modulation of lipid metabolism is a regulatory mechanism for probiotics. Through qPCR, the active role of de novo fatty acid synthesis in the transcriptional upregulation of *E. coli* + glucose treated worms was shown. However, *P. acidilactici* MNL5 + glucose-treated worms showed that transcription repression in the fatty acid desaturase genes is decreased. Therefore, we have shown that *P. acidilactici* MNL5 fat 4, fat 5, and fat 6 are regulated via a mechanism that suppress lipid accumulation through the *C. elegans* (DAF-2 and LIU1) model.

## 4. Materials and Methods

### 4.1. Isolation and Characterization of Cholesterol Reducing LAB

LAB isolated from fermented herbal medicine [MNL–Moola nivarana lehyam, SKML—Solaimalai Kandan Kathri Lehiyam, SKSPL–Sarkaraikolli special Lehiyam, TVL—Thuthuvalai Lehyam, GSC—Gastone Lehiyam] collected from Vadalur Arutjothi Vaithiyasalai [P] Limited, Tiruchirappalli, Tamil Nadu, India. Moola nivarana lehyam: the primary herbal plant is *Withania somnifera*, which belongs to the Solanaceae family and is known as Indian Ginseng or Ashwagandha and is known in Ayurvedic medicine as MedhyaRasayan (Nootropic herb), *Terminalia chebula*: 3.10%, *Emblica officinalis*: 3.10%, *Terminalia Bellirica*: 3.10%, *Withania somnifera*: 6.85%, *Zingiber officinale* (3.6%), *Phytophthora megasperma* (root rot) (3.6%), *Cyperus rotundus* (7%), *Tamarindus indica* (7%), *Aloe barbadensis miller* (7%), *Cissus quadrangularis* (7%), *Amorphophallus paeoniifolius* (3%), *Arenga pinnata* (5%), butter (10%) mixed with Honey, further naturally fermented at 30 °C for 72 h. Colony morphology, staining, motility, and other biochemical tests were conducted on the isolates. Screening of LAB strains from fermented herbal medicines for cholesterol removal was carried out by determination of BSH activity and cholesterol assimilation.

### 4.2. BSH Activityand Cholesterol Assimilation

For BSH activity measurement, 10 mL of culture grown in MRS broth was spotted onto BSH screening medium, which consisted of de Man Rogosa and Sharpe (MRS; Difco, Sparks, MD, USA) agar supplemented with 0.5% (*w*/*v*) sodium salt taurodeoxycholic acid (TDCA, Sigma Aldrich, St. Louis, MO, USA) and 0.37 g/L of CaCl_2_ [36]. Plates were incubated anaerobically at 37 °C using GasPak EZ anaerobe container systems (Becton, Dickinson and Company, Sparks, MD, USA), after which BSH activity was determined by measuring the diameters of precipitation zones. BSH activity was expressed based on the diameters of precipitation zones on BSH screening medium:-no precipitation; +, precipitation zone of up to 10 mm; ++, precipitation zone of up to 15 mm; and +++, precipitation zone of up to 20 mm.

Cholesterol assimilation was determined using the method of Rudel and Morris [37]. LAB cells grown overnight were inoculated (1%) and cultivated anaerobically using GasPak EZ anaerobe container systems (Becton, Dickinson and Company) at 37 °C in MRS broth supplemented with 0.5% (*w*/*v*) oxgall (Sigma Aldrich, St. Louis, MO, USA) and 0.1 g/L of water-soluble cholesterol (Sigma Aldrich). Following incubation, cells were harvested (9950× *g*, 5 min, 4 °C), after which 1 mL of supernatant was added to 2 mL of 33% (*w*/*v*) potassium hydroxide and 3 mL of 95% (*v*/*v*) ethanol. The mixture was shaken well for 1 min and then heated for 10 min in a 60 °C water bath. After cooling in cold water, 5 mL of hexane was added, followed by mixing and addition of 1 mL of distilled water. The tube was allowed to stand for 10 min at room temperature for phase separation, after which 3 mL of the hexane phase was transferred to a clean tube. The hexane phase was then evaporated under a nitrogen stream. The concentrated phase was added to 4 mL of freshly prepared O-phthalaldehyde (Sigma-Aldrich, 0.5 mg of O-phthalaldehyde/mL of acetic acid), mixed, and permitted to stand at room temperature for 10 min. Following the addition of 2 mL of concentrated sulfuric acid and incubation for 10 min, the absorbance at 550 nm was read using a spectrophotometer (Amersham Biosciences, Uppsala, Sweden). Absorbance values were compared to those obtained using cholesterol standard.

### 4.3. Molecular Identification of LAB

Using universal primer, 16S rRNA (27F-5′ AGA GTT TGA TCM TGG CTC AG 3′ and 1492R-5′ GGT TAC CTT GTT ACG ACT 3′) for bacteria, the genotype and species level identification was performed at Macrogen, Seoul, Korea. Entirely attained gnomic sequences were analyzed using the Basic Local Alignment Search Tool (BLAST) and further the sequences were entered in Gen Bank, NCBI (Bankit). All cultures were incubated at 37 °C for 24 h on MRS agar.

### 4.4. Safety Aspect of LAB

#### 4.4.1. Antibiotic Susceptibility

Isolated strains were tested against 9 antibiotics (30 µg L^−1^) with different modes of actions [38] were tested against Gentamicin (Gen), Imipenem (Imp); Erythromycin (Ery), Novobiocin (Nov), Tetracycline (Tet), Clindamycin (Cli), Meropenem (Mer), Ampicillin (Amp), and Penicillin (Pen) based on disc diffusion method. The nutrient agar was scrutinized for the diameter of the inhibition zone was measured with calipers at 37 °C for 24 h. [38]. All antibiotics were obtained (Sigma Aldrich, St. Louis, MO, USA) and tested against LAB using a disc diffusion method. After 24 h incubation, agar plates have been evaluated for the presence or absence of an inhibition zone at 37 °C.

#### 4.4.2. Antibacterial Activity

The antibacterial activity was evaluated using the LAB cell-free supernatants by the agar well diffusion method. All isolates screened were cultivated in MRS Broth and incubated 24 h at 37 °C. To extract the bacterial supernatant, a culture centrifugation with isolates was spin for 25 min at 4 °C, using 6000× *g* [3,39]. These human pathogens, *Escherichia coli* ATCC 35150, *Staphylococcus aureus* ATCC 13,150, *H. pylori* ATCC 43,504, *Bacillus cereus* ATCC 14,579 and *Candida albicans* KCTC 7965 were obtained from the American Type Culture Collection (ATCC, Manassas, VA, USA) and Korean Collection for Type Cultures (KCTC, Seoul, Korea).

#### 4.4.3. Hemolytic Activity

The hemolytic activity was evaluated using the method described earlier [40]. Briefly, the isolate was streaked on nutrient agar plate supplemented with 5% goat blood and incubated for 24 h–48 h at 37 °C to detect patterns of hemolytic activity.

### 4.5. Gastrointestinal Survival under In Vitro Conditions

#### 4.5.1. Tolerance to Low pH

The tolerance to low pH of selected lactic acid bacterial strains was determined using the method described by Mokoena [41], with some modifications. Briefly, the active strains grown in MRS broth were inoculated (1.5%) in 10 mL of fresh MRS broth, incubated at 37 °C for 24 h and then centrifuged at 6000× *g* for 15 min. The pellets were suspended in sterile phosphate-buffered saline (PBS; Gibco™, Thermo Fisher Scientific, Waltham, MA, USA) containing 9 g/L NaCl (Sigma Aldrich, St. Louis, MO, USA), 9 g/L Na_2_HPO_4_⋅2H_2_O and 1.5 g/L KH_2_PO_4_ adjusted to a pH of 2.5. The mixture was incubated at 37 °C for 4 h, and samples were taken at time 0 and after 4 h. These samples were serially diluted in sterile saline (0.85% NaCl) solution and plated on MRS agar; the viable cells were determined after incubation at 37 °C for 24 h, and the percentage of survival of the bacteria was calculated as follows: % survival = [CFU of viable cells survived/CFU of initial viable cells inoculated] × 100.

#### 4.5.2. Tolerance to Pepsin

To test the bacterial tolerance to pepsin, a simulated gastric juice was prepared by adding 3 mg/mL pepsin (Sigma Aldrich, St. Louis, MO, USA) to sterile saline solution (0.85% NaCl, *w*/*v*) and adjusting the pH to 2.5. The fluid was inoculated with active cultures at an inoculum size of 1% (*v*/*v*) and incubated at 37 °C for 4 h. The viable cells were determined at an initial (T1) and final incubation time (T2) by the spread plate method by Oh et al. [42]. The percentage survival of the bacteria was calculated as follows: % survival = [CFU of viable cells survived/CFU of initial viable cells inoculated × 100] percentage of survival [3,42].

#### 4.5.3. Resistance of Pancreatin

The resistance of pancreatin was tested as described by Oh et al. [42], with some modifications [3]. Briefly, 0.3% (*w*/*v*) bile salts (Sigma Aldrich, St. Louis, MO, USA) and 1 mg/mL pancreatin (Sigma Aldrich, St. Louis, MO, USA) were dissolved in sterile saline solution (0.85% NaCl, *w*/*v*) adjusted to pH 8.0. The fluid was inoculated with 1% (*v*/*v*) LAB cultures and incubated at 37 °C for 6 h. The viable cells were determined before and after incubation by the spread plate method. The percentage survival of the bacteria was calculated as follows: % survival = [CFU of survivors/CFU of inoculums × 100].

### 4.6. In Vivo Studies

#### 4.6.1. Maintain the *C. elegans*

In the *Caenorhabditis elegans*, the gene daf-2 encodes the insulin-like growth factor 1 receptor (IGF-1). The *C. elegans* (DAF-2(e1370) and LIU1 (ldrIs1) model) are obtained from the Caenorhabditis Genetics Center (CGC), Minnesota. Then *E. coli* OP50 was placed in the NGM agar plates to supplement the arrested harmonized L1 worm population.

#### 4.6.2. *C. elegans* Synchronizing

The young adult worms hatched from eggs were treated with sterile water containing 0.5 M NaOH and 0.5% bleaching solution (sodium hypochlorite) for synchronization. After vortex mixing at 2 min intervals, worms were washed three times in M9 buffer (3.0 g KH_2_PO_4_, 6.0 g Na_2_HPO_4_, 0.5 g NaCl, 1.0 g NH_4_Cl, H_2_O/1 L, sterilized at 121 °C) via centrifugation (1200× *g* for 200 s). To prepare worms for all assays, approximately 3000 synchronized eggs were hatched in M9 buffer at 20 °C overnight. The L1 larvae were subsequently transferred to nematode growth medium (NGM) agar with a lawn of *E. coli* OP50 and incubated at 25 °C for 48 h to reach the L4 stage. The test was performed by Wong et al. [43] method.

#### 4.6.3. Chemotaxis Assay

The chemotaxis test method was followed by Margie et al. [44]. Thirty L4 stage *C. elegans* worms were administered MNL5 and OP50 diet include Petri plates. To further examine the response *C. elegans* has to a glucose diet; we have established an experiment to determine whether animals distinguish between a spot of food that is the standard OP50 *E. coli* diet and a spot of MNL5 mixed with glucose. At 1 h following the assays, the attractive and counter-attracting paths have been identified as the number of animals. For each condition, triplicate chemotaxis analyses were performed. For each experiment, a chemotaxis index (CI) was calculated using the following formula: Chemotaxis index (CI) = [number of worms in the LAB bacteria strain–number of worms in OP50 (control)/total number of worms applied]. A + 1.0 score indicates maximal attraction towards the target and represents 100% of the worms arriving in the quadrants containing the chemical target. An index of −1.0 is evidence of maximal repulsion.

#### 4.6.4. Establishment of Glucose Diet with LAB

*C. elegans* are used for a synchronized age test, and the experiment was developed in fluid M9 buffer, and eggs are treated with sodium hypochlorite. Our study aimed at achieving glucose focus in a concentration of 10–15 mmol/L *C. elegans* in the entire body. The 50 μL of glucose (18 mg) diet with LAB, which is seeded at 37 °C at 4 h with NGM plates. L4 larvae are moved to the nematode growth medium after four hours.

#### 4.6.5. Bacterial Colonization Assay

A bacterial colonization analysis was conducted for the *C. elegans* gut colonization of *P. acidilactici* MNL5, led by Antunes et al. [45]. On the first day, lactic acid bacteria were seeded on NGM plates containing 100 worms. On the 3rd, 5th, 7th, 9th and 2th day of incubation, the total number of colonies present in the gut of the young adult (L4) stage worm was determined using the following procedure: Five worms were washed in 5 μL drops of M9 buffer to inhibit pharyngeal pumping and expulsion. The washed nematodes were placed in a 1.5 mL centrifugation tube containing 50 μL of PBS buffer with 1% Triton X-100 and mechanically disrupted using a mortar and pestle. Worm lysate was diluted into PBS and plated on MRS agar at a temperature of 37 °C. LAB colonies have been enumerated for the calculation of nematode-bacteria. Compared to *E. coli* with lactic acid bacteria, the procedure was followed Bito et al. [46].

#### 4.6.6. Lifespan Assay

*C. elegans* longevity was measured using the Smolentsevae et al. [47] method already explained. In the test, 50 μL of the TS and MRS broth contain bacteria (OP50-Control and MNL5- test) that were seeded over a 35 mm NGM plate (5-fluorodeoxyuridine FUdR 40 mM for prevents the reproduction) in 24 h at 37 °C. The 100 L4 stage *C. elegans* have been incubated at 20 °C and counted every 24 h as dead worms. Every three days, living worms were moved to fresh NGM plates with bacterial target strain. To compare the effects of glucose against MNL5, first worms with OP50 were cultivated and, the mean lifespan were determined by Chelliah et al. [3]. The three experiments for each strain were performed.

#### 4.6.7. Body Size Measurements

The Olympus SZ 61 zoom stereomicroscope connected to HK3.1 CMOS Camera was used to photograph individual animals after the 0 and 15th day. The size of a worm body measured through a central nematode axis has been assessed by the use of ToupViewTM 3.7 software after cultivating. On at least three independent experiments, at least five nematodes were pictured.

#### 4.6.8. Nile Red and Oil Red O (ORO) Staining Methods

The lipid reduction studies using Nile red and Oil Red O staining methods. The L4 worms used in this study were washed by liquid M9 medium and treated by sodium hypochlorite treatment. Then drop of Nile Red (0.05 μg/mL) solution were added to the worms, which were then incubated for 30 min, washed with 25% ethanol twice, and photographed in a Fluorescence microscope (Olympus CKX53, Tokyo, Japan) analysis. The fresh ORO solution was prepared by diluting stock (0.5% ORO in isopropanol) to a 60% solution with water, filtering (0.45 µM filter), stirring at room temperature overnight, and filtering again just before use. Afterwards, L4 worms were collected, washed, and fixed in 60% isopropanol for 5 min. Then, the fixed worms were incubated in the ORO solution for 6 h in a wet chamber with gentle shaking in the dark, washed with PBS, and mounted on a 2% agarose pad for microscopy visualization [48]. All of the worms were photographed with same setups and same exposure times.

#### 4.6.9. Fluorescence Quantification

The fluorescence intensity was quantified by Image J software. Nile red and Oil Red O stained lipids were predominantly found in the gut, and the anterior region was significantly brighter than the posterior. The region for integrating fluorescence density measurements was chosen from the intestine’s anterior front to vulva. Six L4 worms were randomLy chosen for analysis. For Image J analysis, pictures were captured at a magnification of 20×.

#### 4.6.10. RNA Extraction and Primer Deign

The gene expression of *C. elegans* (DAF-2& LIU1) has been analyzed in *E. coli* OP50, *E. coli* OP50 + glucose and *P. acidilactici* MNL5+glucose, fed worm populations. Synchronized populations were obtained under specific feed conditions from embryos isolated with gravy adults. After feeding time (12 days of 50% of population), worms were obtained with an M9 buffer and washed three times in Eppendorf tubes for disturbed by sonication (3 pulses at 10 W, 20 s/pulse) [49]. Following the manufacturer’s instructions with certain modifications, Total RNA has been extracted from nematodes using RNAiso Plus (Takara Co., Ltd., Kusatsu, Japan). RNA quantification was performed with the Eppendorf AG22331 Biospectrophotometer (Eppendorf AG, Hamburg, Germany). 260/280 and 260/230 absorption ratios assessed RNA purity.

Primer design Specific primers for the differentiation of lipids targeted on selective genes metabolism is involved in the regulation of DAF-2/insulin signaling. The respective primers (fat-4, fat-5, fat-6) were designed using Primer Express^®^ Software v3.0.1, (Applied Biosystems; Foster City, CA, USA) and were prepared as described by Bioneer Corporation (Daejeon, Korea). The primers used in the study along with their primer sequence and product size are shown in Appendix A. In addition, to enhance the detection range of the primers, the International Union of Pure and Applied Chemistry (IUPAC, Research Triangle, NC, USA) was used for confirmation. The primers were designed to keep the melting temperature (Tm) of the PCR amplicons between 70 and 85 °C and each amplicon Tm was separated by approximately 2 °C (Appendix A, Appendix A). All amplicons were predicted using the Bio-Edit Software [50] and by applying Primer-BLAST tool and Tm, specificity of the designed primers was determined [14].

#### 4.6.11. Quantitative PCR Analyses

All qPCR assays were performed using StepOne™ real-time PCR System (Applied Biosystems, Foster City, CA, USA). For each reaction, 10 μL of MeltDoctor™ HRM Master Mix, 2 μL of genomic DNA (10 ng μL^−1^), 0.5 μL of each primer (10 pg μL^−1^) and 7 μL of double distilled water were added. Gene expression differences between treated and untreated worms were quantified using the relative quantification 2^−∆∆Ct^ method [51].

### 4.7. Statistical Analysis

Reproducibility of the qPCR was determined with three independent experiments conducted in duplicates. Data collected from quantitative tests (acid tolerance, pepsin & pancreatin, antibacterial activity, antibiotic susceptibility, and cholesterol assimilation), further the results were The data obtained from each experiment was expressed as mean and standard deviation by Microsoft Office Excel 2010 (Microsoft Corporation, Redmond, Washington, DC, USA).

## 5. Conclusions

The present work reveals that fermented medicines contain probiotic bacteria. As such, their use may lead decreased cholesterol concentrations. Additionally, it was found that *P. acidilactici* MNL5 bacteria have better survival ability in the GI tract. Our results showed longer lifespans for *P. acidilactici* MNL5 and shorter colonization for *C. elegans*. The transcription of genes encoding fatty acid desaturase was significantly suppressed in *P. acidilactici* MNL5 treated worms, according to our qPCR results. These results confirm that *P. acidilactici* MNL5 is a viable approach to lipid reduction, supplementing dietary efforts for lipid reduction in the DAF-2 and LIU1 model for obesity management. In addition, the information presented in this paper will support future studies on mice obesity models.

## Figures and Tables

**Figure 1 ijms-23-01276-f001:**
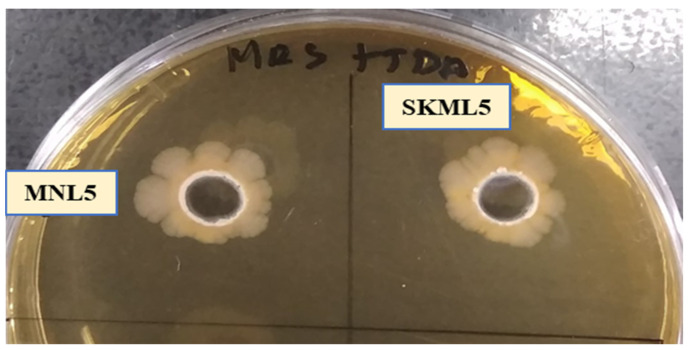
(The BSH activity of LAB isolates grown on bile salt–MRS medium as manifested by the formation of precipitation zone around the colony.

**Figure 2 ijms-23-01276-f002:**
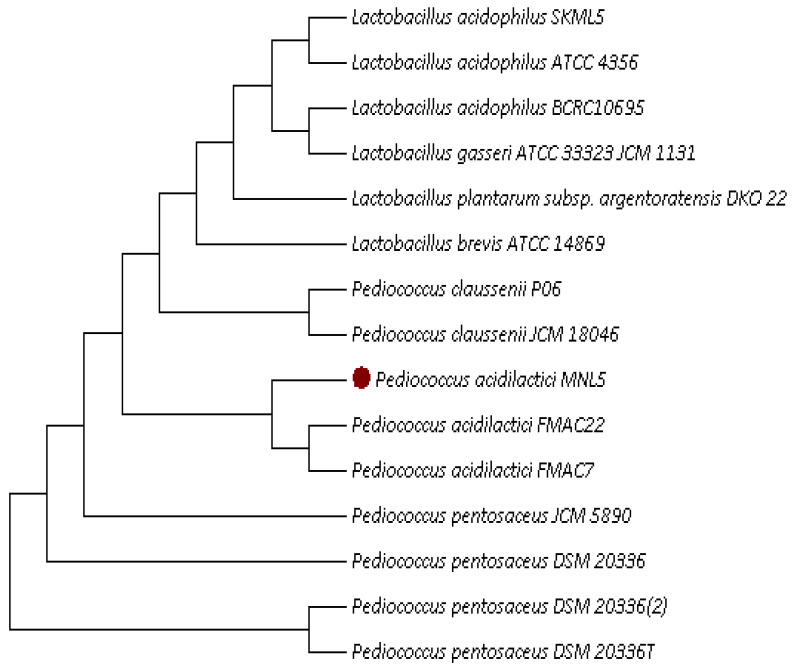
The 16S rRNA sequence of *Pediococcus acidilacticii* MNL5 based phylogenetic tree was constructed by the neighbor-joining method. The numbers at the nodes are bootstrap values based on 0.0001 replications.

**Figure 3 ijms-23-01276-f003:**
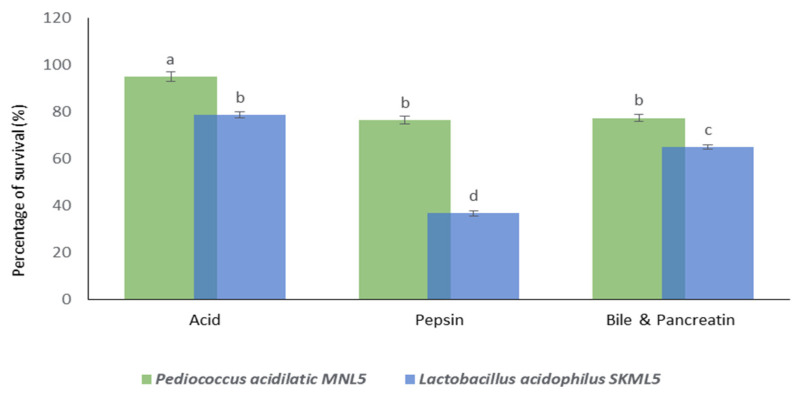
Stability of Lactic acid bacteria towards tolerance of acid, pepsin and Bile & Pancreatin. Values are expressed as the mean ± standard deviation (*n* = 3). Different superscripts (^a, b, c, d^) represent significantly different values (*p* < 0.05).

**Figure 4 ijms-23-01276-f004:**
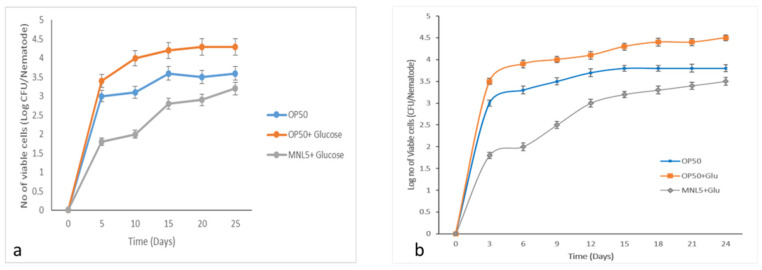
Bacterial colonization in *C. elegans*. (**a**) DAF-2, (**b**) LIU1 model; the treatments such as, OP50 (Control), MNL+ GLU (*P. acidilactici* MNL5 with Glucose) [Treatment]), OP50+GLU (*E. coli* OP50 with Glucose [Positive Control]; Evaluation of MNL5 colonization (numbers of log CFU/nematode) in the nematode intestine. In each experiment, 100 worms were administered, repeated twice and plotted using OASIS II. All results are presented as the means ± standard error mean.

**Figure 5 ijms-23-01276-f005:**
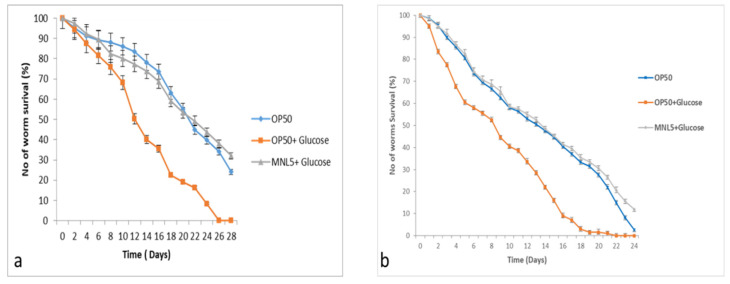
Glucose supplements diet by lifespan of *C. elegans*. (**a**) DAF-2; (**b**) LIU1 model. The treatments such as, OP50 (Control), MNL + GLU (*P. acidilactici* MNL5 with Glucose) [Treatment]), OP50+GLU (*E. coli* OP50 with Glucose [Positive Control]; in each experiment, 100 worms were administered, repeated twice and plotted using OASIS II. All results are presented as the means ± standard error mean.

**Figure 6 ijms-23-01276-f006:**
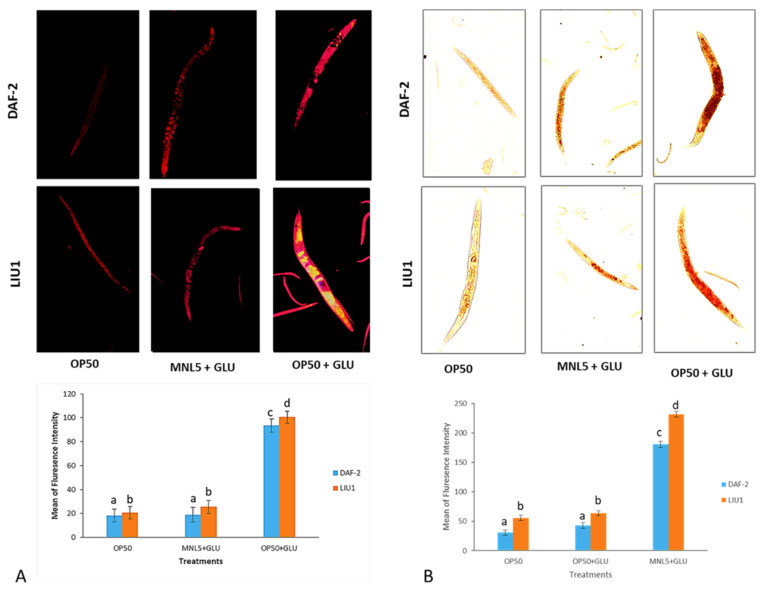
Lipid content visualized by *C. elegans* body (DAF-2 and LIU1 model) by fluorescence intensity measured by ImageJ software. Microscopic images showing different treatments such as OP50 (Control), MNL+ GLU (*P. acidilactici* MNL5 with Glucose) [Treatment]), OP50+GLU (*E.coli* OP50 with Glucose [Positive Control]; (**A**) Nile Red staining by fluorescence microscope (20× magnification); (**B**) Oil Red O staining by light microscope (20× magnification). Bar diagram presents the relative lipid content of worms. The error bar represents the mean ± SEM). Different superscripts (^a, b, c, d^) represent significantly different values (*p* < 0.05).

**Figure 7 ijms-23-01276-f007:**
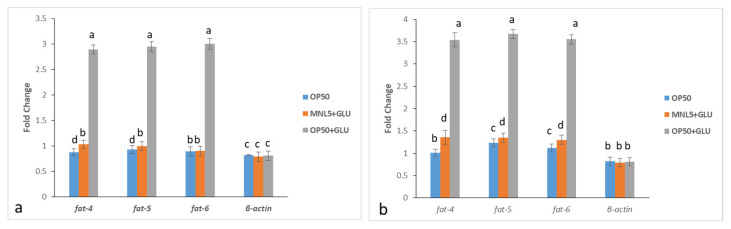
The qPCR fold change in expression pattern of fat genes. (**a**) DAF-2; (**b**) LIU1; The bar diagram indicates the following changes in worms exposed different diets to gene expression: OP50; MNL5+ Glucose; OP50+ Glucose. Values are expressed in mean ± standard deviation (*n* = 3). Different superscripts (^a, b, c, d^) represent significantly different values (*p* < 0.05).

**Table 1 ijms-23-01276-t001:** Assimilation of cholesterol, Haemolytic activity and BSH of isolated LAB strains from fermented herbal medicine.

No	Strain Name	Colour Morphology	Shape	Hemolytic Activity	BSH Activity	Cholesterol Assimilation (%)
1	MNL 1	White/Smooth	cocci	NA	++	25.3 ± 0.54 ^g^
2	MNL 2	White	rod	NA	+	18.5 ± 0.41 ^h^
3	MNL 3	White/Round	rod	NA	-	10.2 ± 0.39 ^h^
4	MNL 4	White/Smooth/Round	rod	NA	-	11.7 ± 0.24 ^h^
5	**MNL 5**	**White/Smooth**	**cocci**	**NA**	**+++**	**89.5 ± 1.09 ^a^**
6	MNL 6	White	rod	NA	-	25.5 ± 0.44 ^g^
7	MNL 8	White	cocci	NA	-	23.4 ± 0.38 ^g^
8	MNL 9	White	cocci	NA	+	31.5 ± 0.61 ^f^
9	MNL 10	White/Round	rod	NA	+	24.5 ± 0.50 ^g^
10	MNL 11	White/Smooth/Round	short rod	NA	++	47.5 ± 0.36 ^e^
11	GSC 5	White/Smooth	rod	NA	+	31.0 ± 0.37 ^f^
12	GSC 8	White	rod	NA	-	12.4 ± 0.47 ^h^
13	GSC 10	White	short rod	NA	+	10.5 ± 0.39 ^h^
14	TVL 11	White/Smooth	short rod	NA	+	14.6 ± 0.40 ^h^
15	TVL 12	White	short rod	NA	-	12.4 ± 0.55 ^h^
16	SKML 3	White/Round	rod	NA	+	22.5 ± 0.51 ^g^
17	SKML 4	White	rod	NA	-	9.5 ± 0.64 ^h^
18	SKML 5	White/Smooth	cocci	NA	+++	77.1 ± 0.98 ^b^
19	SKML 6	White/Smooth	rod	NA	++	69.5 ± 0.78 ^c^
20	SKML 8	White/Smooth/Irregular	rod	NA	+	53.43 ± 0.63 ^d^
21	SKML 10	White	cocci	NA	+	47.56 ± 0.51 ^e^
22	SKML 11	White	short rod	NA	+	45.21 ± 0.69 ^e^
23	SKML 12	White	cocci	NA	-	25.80 ± 0.40 ^g^
24	SKML 13	White	short rod	NA	-	22.4 ± 0.35 ^g^
25	SK SPL 1	White/Rough	short rod	NA	-	21.4 ± 0.44 ^g^
26	SK SPL 6	White	cocci	NA	++	57.4 ± 0.64 ^a^
27	SK SPL7	White/Smooth	cocci	NA	+	42.5 ± 0.51 ^e^
28	GSC 1	White	cocci	NA	+	38.4 ± 0.38 ^f^
29	GSC 11	White	short rod	NA	+	32.5 ± 0.33 ^f^
30	GSC 12	White	cocci	NA	++	41.5 ± 0.46 ^e^
31	SKML 18	White	short rod	NA	++	57.8 ± 0.48 ^d^
32	SKML 20	White/Smooth	cocci	NA	++	68.6 ± 0.64 ^c^

-, no precipitation; +, precipitation zone up to 10 mm; ++, precipitation zone up to 15 mm; and +++, precipitation zone up to 20 mm. NA-γ- haemolysis (no zones around colonies). Values are expressed in mean ± standard deviation (*n* = 3) Different superscripts (^a, b, c, d, e, f, g, h^) represent significantly different values (*p* < 0.05).

## Data Availability

All data generated for this study are available from the corresponding authors upon reasonable request.

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
