# Peer review of "Anti-Obesity Efficacy of Pediococcus acidilactici MNL5 in Canorhabditis elegans Gut Model"

_ijms, 2022, doi:10.3390/ijms23031276_

Round 1

Reviewer 1 Report

In my opinion all remarks were addressed. For this reason, I can recommend this article for publication.

Author Response

Reviewer response 1

Manuscript ID: ijms- 1557478

Title: Anti-obesity efficacy of Pediococcus acidilactici MNL5 in Caenorhabditis elegans gut model

In my opinion, all remarks were addressed. For this reason, I can recommend this article for publication.

We are grateful to the reviewer for providing valuable comments on the article and also for your precious time. As per the reviewer's valuable comments, the manuscript has been edited by a native English speaker.

Reviewer 2 Report

The author investigated isolated from fermented Indian herbal medicine. In comparison to other strains, MNL5 had stronger Bile salt hydrolase (BSH), Cholesterol-lowering ability, probiotic properties and antibacterial activity. The author investigates to in vivo model of C. elegans. This experiment P. acidilactici MNL5 supplementation to increase life span (DAF-2 & LIU1 model) (P < 0.05), and, they checked lipid droplets in the C. elegans (fluorescent microscope) and expression of fat genes related to obesity phenotypes. The Manuscript is interesting results to probiotic strain have anti-obesity efficacy. The experiments have been well planned, and obtained results are well documented by data presented in the main body manuscript and supplementary materials. However, I would like to ask for some minor comments.

Minor comments:

Inline no 43 Lactobacillus and Pediococcus are maintaining the intestinal microbial ecosystem and impact on reduced function of cholesterol [9, 10]. The genus name should be in italics.

Line no 73 Withania Somnifera: 6.85%. The species name should be in italics.

Line no 85, 95, and 96. It should be “Sigma-Aldrich”;

Line no 111. Include the primer sequence incorporate in this section.

Line no 176 “0.5 M NaOH and 0.5% bleaching solution (sodium hypochlorite and sodium hydroxide). In closed bracket should be remove sodium hydroxide. Already author is mentioned NaOH.

Subsection 2.7.5; 2.7.6 Include the worm number for this experiment. Author mentioned 100 worms used in this study showed in Figures 4, 5. Kindly incorporate the materials method section.

Subsection 2.7.9, Line no 250-252- Language and style correction required. For measurement, six L4 worms were selected at random. Images were captured at 20x magnification for Image J analysis.

Subsection 2.8. Kindly include the statistical analysis tools and version.

Line no 337-339- Language and style correction required, Above all, MNL5 exhibits substantial inhibitory activity against Escherichia coli, Staphylococcus aureus, Bacillus cereus and H. pylori while lesser activity was observed with Candida albicans KCTC 7965.

Line no 364- Language and style correction required. Based on these results, MNL5 is obvious to act as an improvement in growth when used as nematode supplements.

Line no 376, microbe’sobes name should be in italics, Language and style correction required. In contrast, when administered MNL5+Glucose and OP50, the amount of gut bacteria accumulated is less when compared to E. coli OP50 + Glucose colonization, which is increased (Figure 4).

Line no 430-432- Language and style correction required. The pathway for this experiment is to indicate that daf-2 & LIU1 responds to the absence of either lipid through up-regulating pathways of fatty acid synthesis and fatty acid downregulation pathways.

Line no. 436, it should be in fat-6

Line no 486-488- Language and style correction required. During our study, the life span of daf-2 decreased considerably due to lipid metabolism. P. acidilactici MNL5, however, improved the daf-2 model's lifespan.

Line no 515 Language and style correction required. The author is performed QPCR or RT-PCR, should be change. Based on the RT-PCR suggestion, treatment with P. acidilactici MNL5 worms show that the transcription of the genes of fatty acid desaturase is downregulated.

Line no 521. Language and style correction required.  The results confirm that P. acidilactici MNL5 is a viable approach to lipid reduction, supplementing dietary efforts for lipid reduction in daf-2 model by obesity management. In 522 addition, the data support future studies on the mice model for obesity.

In reference section according the Journal format. The author should be mention the genus and species name should be in italics.

Author Response

Review Report (Reviewer 2)

Manuscript ID: ijms-1557478

Title: Anti-obesity efficacy of Pediococcus acidilactici MNL5 in Caenorhabditis elegans gut model

The author investigated isolated from fermented Indian herbal medicine. In comparison to other strains, MNL5 had stronger Bile salt hydrolase (BSH), Cholesterol-lowering ability, probiotic properties and antibacterial activity. The author investigates to in vivo model of C. elegans. This experiment P. acidilactici MNL5 supplementation to increase life span (DAF-2 & LIU1 model) (P < 0.05), and, they checked lipid droplets in the C. elegans (fluorescent microscope) and expression of fat genes related to obesity phenotypes. The Manuscript is interesting results to probiotic strain have anti-obesity efficacy. The experiments have been well planned, and obtained results are well documented by data presented in the main body manuscript and supplementary materials. However, I would like to ask for some minor comments.

The authors were grateful to the reviewers for providing valuable comments towards making the manuscript in readable and suitable form for the readers.

Minor comments:

Inline no 43 Lactobacillus and Pediococcus are maintaining the intestinal microbial ecosystem and impact on reduced function of cholesterol [9, 10]. The genus name should be in italics.

Lactobacillus and Pediococcus are maintaining the intestinal microbial ecosystem and impact on reduced function of cholesterol [9, 10].

Line no 73 Withania Somnifera: 6.85%. The species name should be in italics.

Moola nivarana lehyam is a fermented herbal medicine, mainly applied for reducing obesity, which consist of combined dried spices such as Medhya Rasayan (Nootropic herb), Terminalia chebula: 3.10%, Emblica officinalis: 3.10%, Terminalia bellirica: 3.10%, Withania somnifera (6.85%), Zingiber officinale (3.6%), Phytophthora megasperma (root rot) (3.6%), Cyperus rotundus (7%), Tamarindus indica (7%), Aloe barbadensis miller (7%), Cissus quadrangularis (7%), Amorphophallus paeoniifolius (3%), Arenga pinnata (5%), butter (10%), where naturally fermented at 30°C for 72 h.

Line no 85, 95, and 96. It should be “Sigma-Aldrich”;

Sigma-Aldrich

Line no 111. Include the primer sequence incorporate in this section.

Using universal primer, 16S rRNA (27F-5ʹ AGA GTT TGA TCM TGG CTC AG 3ʹ and 1492R-5ʹ GGT TAC CTT GTT ACG ACT 3ʹ) for bacteria, the genotype and species level identification was performed at Macrogen, Seoul, South Korea.

Line no 176 “0.5 M NaOH and 0.5% bleaching solution (sodium hypochlorite and sodium hydroxide). In closed bracket should be remove sodium hydroxide. Already author is mentioned NaOH.

The young adult worms hatched from eggs were treated with sterile water containing 0.5 M NaOH and 0.5% bleaching solution (sodium hypochlorite) for synchronization.

Subsection 2.7.5; 2.7.6 Include the worm number for this experiment. Author mentioned 100 worms used in this study showed in Figures 4, 5. Kindly incorporate the materials method section.

A bacterial colonization analysis was conducted for the C. elegans gut colonization of P. acidilactici MNL5, led by Antunes et al [24]. On the first day, lactic acid bacteria were seeded on NGM plates containing 100 worms.

Subsection 2.7.9, Line no 250-252- Language and style correction required. For measurement, six L4 worms were selected at random. Images were captured at 20x magnification for Image J analysis.

Six L4 worms were randomly chosen for analysis. For Image J analysis, pictures were captured at a magnification of 20x.

Subsection 2.8. Kindly include the statistical analysis tools and version.

Data collected from quantitative tests (acid tolerance, pepsin & pancreatin, antibacterial activity, antibiotic susceptibility, and cholesterol assimilation), further the results were The data obtained from each experiment was expressed as mean and standard deviation by Microsoft Office Excel 2010 (Microsoft; USA).

Line no 337-339- Language and style correction required, Above all, MNL5 exhibits substantial inhibitory activity against Escherichia coli, Staphylococcus aureus, Bacillus cereus and H. pylori while lesser activity was observed with Candida albicans KCTC 7965.

Above all LAB, the MNL5 exhibits the substantial inhibitory activity against Escherichia coli, Staphylococcus aureus, Bacillus cereus and H. pylori while lesser activity observed with Candida albicans KCTC 7965 (Sup Table 2).

Line no 364- Language and style correction required. Based on these results, MNL5 is obvious to act as an improvement in growth when used as nematode supplements.

Based on the results, MNL5 appears to promote nematode growth when administered as a nematode supplement.

Line no 376, microbe’sobes name should be in italics, Language and style correction required. In contrast, when administered MNL5+Glucose and OP50, the amount of gut bacteria accumulated is less when compared to E. coli OP50 + Glucose colonization, which is increased (Figure 4).

Figure 4. Bacterial colonization in C. elegans. a) DAF-2; b) LIU1 model). The treatments such as, OP50 (Control), MNL+ GLU (P. acidilactici MNL5 with Glucose) [Treatment]), OP50+GLU (E.coli OP50 with Glucose [Positive Control]; Evaluation of MNL5 colonization (numbers of log CFU/nematode) in the nematode intestine. In each experiment, 100 worms were administered, repeated twice and plotted using OASIS II. All results are presented as the means ± standard error mean.

Line no 430-432- Language and style correction required. The pathway for this experiment is to indicate that daf-2 & LIU1 responds to the absence of either lipid through up-regulating pathways of fatty acid synthesis and fatty acid downregulation pathways.

This experiments pathway is to demonstrate that daf-2 & LIU1 model responds to the absence of either lipid through up-regulating or down-regulating pathways.

Line no. 436, it should be in fat-6

As a result, we showed that fat-4, fat-5, and fat-6 are modulated by a mechanism that suppresses lipid reduction by P. acidilactici MNL5 in a C. elegans fat gene model (Figure 7).

Line no 486-488- Language and style correction required. During our study, the life span of daf-2 decreased considerably due to lipid metabolism. P. acidilactici MNL5, however, improved the daf-2 model's lifespan.

During our study, the life span of DAF-2 and LIU1 model decreased considerably due to lipid metabolism. The longevity of the DAF-2 and LIU1 models significantly improved by P. acidilactici MNL 5.

Line no 515 Language and style correction required. The author is performed QPCR or RT-PCR, should be change. Based on the RT-PCR suggestion, treatment with P. acidilactici MNL5 worms show that the transcription of the genes of fatty acid desaturase is down regulated.

Meanwhile, our results extended the lifespan of P. acidilactici MNL5 and diminished C. elegans colonization. The P. acidilactici MNL5 strongly suggested a fluorescence analysis to reduce the fat accumulation and lipid droplets in C. elegans experiments.

Line no 521. Language and style correction are required.  The results confirm that P. acidilactici MNL5 is a viable approach to lipid reduction, supplementing dietary efforts for lipid reduction in daf-2 model by obesity management. In 522 addition, the data support future studies on the mice model for obesity.

The results confirm that P. acidilactici MNL5 is a viable approach to lipid reduction, supplementing dietary efforts for lipid reduction in DAF-2& LIU1 model by obesity management. In addition, the information supports future studies on the mice model for obesity.

In reference section according the Journal format. The author should be mention the genus and species name should be in italics.

The reference section was corrected according to journal guidelines.

We are grateful to the reviewer for providing valuable comments on the article and also for your precious time.
